# Procaine Abrogates the Epithelial-Mesenchymal Transition Process through Modulating c-Met Phosphorylation in Hepatocellular Carcinoma

**DOI:** 10.3390/cancers14204978

**Published:** 2022-10-11

**Authors:** Min Hee Yang, Chakrabhavi Dhananjaya Mohan, Amudha Deivasigamani, Arunachalam Chinnathambi, Sulaiman Ali Alharbi, Kanchugarakoppal S. Rangappa, Sang Hoon Jung, Hyejin Ko, Kam Man Hui, Gautam Sethi, Kwang Seok Ahn

**Affiliations:** 1KHU-KIST Department of Converging Science and Technology, Kyung Hee University, Seoul 02447, Korea; 2Natural Products Research Center, Korea Institute of Science and Technology (KIST), Gangneung 25451, Korea; 3Department of Studies in Molecular Biology, University of Mysore, Manasagangotri, Mysore 570006, India; 4Division of Cellular and Molecular Research, Humphrey Oei Institute of Cancer Research, National Cancer Centre Singapore, Singapore 169610, Singapore; 5Department of Botany and Microbiology, College of Science, King Saud University, Riyadh 11451, Saudi Arabia; 6Institution of Excellence, Vijnana Bhavan, University of Mysore, Manasagangotri, Mysore 570006, India; 7Cancer and Stem Cell Biology Program, Duke-NUS Medical School, Singapore 169857, Singapore; 8Department of Biochemistry, Yong Loo Lin School of Medicine, National University of Singapore, Singapore 119077, Singapore; 9Department of Pharmacology, Yong Loo Lin School of Medicine, National University of Singapore, Singapore 117600, Singapore; 10Department of Science in Korean Medicine, Kyung Hee University, 24 Kyungheedae-ro, Dongdaemun-gu, Seoul 02447, Korea

**Keywords:** procaine, epithelial-mesenchymal transition, c-Met, hepatocellular carcinoma, orthotopic, invasion, migration

## Abstract

**Simple Summary:**

Epithelial-mesenchymal transition (EMT) is a vital process that leads to the dissemination of tumor cells to distant organs and promotes cancer progression. Aberrant activation of c-Met has been positively correlated with tumor metastasis in hepatocellular carcinoma (HCC). In this report, we have demonstrated the suppressive effect of procaine on the EMT process through the blockade of the c-Met signaling pathway. Procaine downregulated mesenchymal markers and upregulated epithelial markers. Functionally, procaine abrogated cellular migration and invasion. Moreover, procaine suppressed c-Met and its downstream signaling events in HCC models. We report that procaine can function as a novel inhibitor of the EMT process and c-Met-dependent signaling cascades. These results support the consideration of procaine being tested as a potential anti-metastatic agent.

**Abstract:**

EMT is a critical cellular phenomenon that promotes tumor invasion and metastasis. Procaine is a local anesthetic agent used in oral surgeries and as an inhibitor of DNA methylation in some types of cancers. In this study, we have investigated whether procaine can inhibit the EMT process in HCC cells and the preclinical model. Procaine suppressed the expression of diverse mesenchymal markers but induced the levels of epithelial markers such as E-cadherin and occludin in HGF-stimulated cells. Procaine also significantly reduced the invasion and migration of HCC cells. Moreover, procaine inhibited HGF-induced c-Met and its downstream oncogenic pathways, such as PI3K/Akt/mTOR and MEK/ERK. Additionally, procaine decreased the tumor burden in the HCC mouse model and abrogated lung metastasis. Overall, our study suggests that procaine may inhibit the EMT process through the modulation of a c-Met signaling pathway.

## 1. Introduction

Metastasis is an important process in which tumor cells move from one organ to the other, which poses the most life-threatening risk for cancer patients [1,2,3]. Over 90% of cancer deaths are due to metastatic cancer rather than non-metastatic tumors [4,5,6,7]. Epithelial-mesenchymal transition (EMT) leads to the loss of epithelial phenotype and enables the cell to gain mesenchymal features [8,9,10]. During the process of EMT, the expression of epithelial markers (E-cadherin) is lost in cancer cells with a parallel increase in the levels of various mesenchymal markers [11,12,13,14]. EMT can also elevate tumor cell invasion, migration, and metastasis [15,16]. Many growth regulators including transforming growth factor-β (TGFβ), hepatocyte growth factor (HGF) etc. bind to their corresponding receptors on the cell membrane and relay the signals to upregulate the expression of EMT-related transcription factors [17].

EMT is regulated by many signaling cascades including the c-Met signaling pathway [18]. c-Met can be activated by HGF and its activation upon ligand binding can lead to phosphorylation of downstream PI3K/Akt/mTOR and MAPK signaling pathways [18,19]. This phenomenon triggers a variety of biological processes such as metastasis, proliferation, invasion, angiogenesis, and EMT [20,21,22]. Persistent activation of c-Met has been observed in various human malignancies including cancers of the liver, colon, lung, gastric system, and breast [23,24]. HGF/c-Met elevation has been associated with metastatic progression in many major human cancers [25]. Frequent recurrence and metastasis are the contributing factors to the dismal prognosis of hepatocellular carcinoma (HCC) [26]. A recent study indicated that curcumin suppressed HGF-induced EMT and angiogenesis by modulating c-Met and PI3K/Akt/mTOR pathways [27], thus indicating that small molecules affecting the c-Met pathway might be an effective strategy in the treatment of HCC through the suppression of EMT.

Procaine is one of the frequently administered local anesthetic drugs mainly used in oral surgeries [28]. Recent studies have demonstrated the antitumor potential of procaine against different cancer cells including those of the breast, liver, lung, and against osteosarcoma [29,30,31,32,33]. Procaine reduced the methylation of the CpG island in breast cancer and hepatoma [29,30]. In addition, Ying et al. reported that procaine suppressed proliferation and migration but enhanced apoptosis in osteosarcoma cells by causing the upregulation of miR-133b. [31]. Li and colleagues found that procaine downregulated the proliferation and migration of colon cancer cells by inhibiting the ERK/MAPK/FAK pathway [32]. Moreover, we have recently reported that procaine attenuated DNA methylation by inhibiting DNA methyltransferase activity and increased the expression of PAX9, which suppressed cell growth, and promoted apoptosis in oral squamous cell carcinoma [33]. The precise effect of procaine on EMT and tumor metastasis has not been reported in preclinical cancer models so far. In this study, we have investigated the impact of procaine on tumor progression in a preclinical orthotopic HCC cancer model. We noted that procaine abrogated the EMT process by modulating the c-Met signaling pathway and suppressed cancer cell mobility in human HCC (HepG2 and HCCLM3) cells. Procaine also repressed the growth of HCC tumors in the mice model and hampered lung metastasis.

## 2. Results

### 2.1. Procaine Alters the Level of Proteins Linked to EMT

The chemical structure of procaine is provided in Figure 1A. The effect of procaine and HGF on the viability of HepG2 and HCCLM3 cells was evaluated by an in vitro cytotoxicity assay. Procaine did not display significant cytotoxicity in both cell lines up to 200 μM (Figure 1B). Procaine and HGF co-treated cells displayed slightly higher cell viability, and the highest cell viability was observed in the group treated with HGF alone. We analyzed the impact of procaine on the expression of EMT-related proteins. HepG2 and HCCLM3 cells were treated with procaine (0, 30, 50, 100, 200 μM) for 24 h. As shown in Appendix A, expressions of MnSOD, Vimentin, and Snail were markedly suppressed, but that of Occludin was significantly induced at 200 μM of procaine. Therefore, we selected a 200 μM dose, which exhibited very low toxicity for the experiments. Procaine downregulated the levels of mesenchymal markers while upregulating epithelial markers, as shown in Figure 1C (Appendix A). In parallel, HGF treatment triggered the expression of Occludin and E-cadherin, and procaine induced the expression of these markers (Figure 1D and Appendix A). The expression of N-cadherin, Snail, and Occludin in uninduced/HGF-induced cells upon procaine treatment was examined using immunocytochemistry analysis. Procaine also attenuated HGF-simulated N-cadherin and Snail levels, with a parallel increase in Occludin levels (Figure 1E).

### 2.2. Procaine Downregulates the Invasive and Migratory Potential of HGF-Induced HCC Cells

The effect of procaine on cell motility was examined using an invasion assay and a wound healing assay. HGF treatment caused the alteration of cell morphology to a spindle-like shape, and these morphological changes were reversed by procaine treatment (Figure 2A). Furthermore, the results of the wound healing assay showed that HGF treatment elevated the migration of HCC cells, and procaine suppressed the migratory potential (Figure 2B). As shown in Figure 2C, HGF enhanced cell invasion, whereas procaine suppressed this activity. Additionally, the activity of MMP-2 and MMP-9 was examined by gelatin zymography upon procaine treatment in the HGF-induced cells. HGF significantly enhanced MMP activity and procaine treatment significantly suppressed MMP-2 and MMP-9 activity (Figure 2D).

### 2.3. Procaine Suppresses Activation of c-Met-Dependent Signaling Events

Next, the impact of procaine on the constitutive/HGF-induced activation of c-MET-dependent signalling pathways was investigated The treatment with HGF induced the posphorylation of c-Met in HepG2 and HCCLM3 cells and procaine suppressed the HGF-induced activation of c-Met without altering the levels of total c-Met (Figure 3A and Appendix A). A similar inhibitory effect was observed in the downstream effectors of c-Met such as PI3K/Akt/mTOR, MEK, JNK, and p38 (Figure 3B–D and Appendix A), thus, indicating that procaine suppressed the c-Met signalling pathway to revert EMT process. When comparing the procaine efficacy in HepG2 and HCCLM3 cells, it was found that the HGF-induced proteins were reduced more effectively in HCCLM3 cells.

### 2.4. Procaine Attenuates Tumorigenesis and Metastasis in Orthotopic HCC Mouse Model

The toxicity and antitumor activity of procaine was examined in mice models. Acute toxicity studies were performed by the intraperitoneal administration of procaine (25 mg/kg or 50 mg/kg) to a different group of animals. Procaine administration did not show substantial alteration in body weight relative to the control animals (Figure 4A). In addition, there are significant alterations noted in levels of aspartate aminotransferase (AST), alanine aminotransferase (ALT), and blood urea nitrogen (BUN) in the serum of procaine-treated animals (Figure 4B). Additionally, for antitumor studies, HCCLM3-Luc cells derived tumors were grown in NCr nude mice and the animals were treated with 25 mg/kg or 50 mg/kg of Procaine intraperitoneally, twice a week, for four weeks. The procaine-administered animals significantly reduced the tumor growth and lung metastases compared with the control group (Figure 4C–E). Thus, these results suggested that procaine treatment significantly suppressed tumorigenesis and metastasis without notable toxicity in the tested animals.

### 2.5. Procaine Alters the Expression of Oncogenic and EMT-Related Proteins in Tumor Tissues

An immunohistochemical analysis was performed to determine the expression of phosphorylated-c-Met, proliferation indicator (Ki-67), and EMT markers (Vimentin and E-cadherin) in tumor tissues derived mice. Procaine significantly suppressed the expression of phospho-c-Met, Ki-67, and Vimentin and elevated the levels of E-cadherin (Figure 5A). To evaluate the impact of procaine on the EMT process, the expression of EMT-related proteins was examined by western blot analysis in tumor tissues derived from control and procaine treated animals. As shown in Figure 5B (Appendix A), the expression of MnSOD, Vimentin, N-cadherin, MMP-2, and Snail were decreased, whereas Occludin and E-cadherin expressions were increased in procaine treated groups. Procaine also significantly reduced the activation of c-Met in procaine-treated groups compared to control animals (Figure 5C and Appendix A). Additionally, the phosphorylation of c-Met downstream signals such as PI3K, Akt, mTOR, MEK, and ERK was reduced in procaine treated groups (Figure 5D and Appendix A). These results were in agreement with our in vitro data.

## 3. Discussion

Procaine is a synthetic compound which can exhibit pleiotropic neoplastic actions against various types of cancers [29,30,31,32,33]. A recent study reported that procaine can display a growth-inhibitory effect and demethylation effects on human hepatoma cells in vitro and in vivo [30]. There are no previous studies deciphering its possible impact on the EMT process. In this study, we have herewith demonstrated that procaine abrogated the EMT process in HCC cells and the preclinical orthotopic cancer model. Our results showed that procaine effectively attenuated the HGF-induced EMT process through down-regulating the expression of mesenchymal markers and up-regulating the expression of epithelial markers. Additionally, procaine substantially mitigated HGF-driven cell motility. The observed effects are primarily mediated through the ability of procaine to interfere with the activation of c-Met and its associated signaling pathways.

EMT is a crucial and complex event that can stimulate diverse signaling events responsible for tumor progression [16,34,35]. During the EMT process, the expression of epithelial and mesenchymal markers is downregulated and upregulated, respectively [36]. Previous studies have reported that HGF regulates the EMT process and accelerates tumorigenesis [27,37]. In an interesting report, curcumin inhibited HGF-driven EMT-associated changes in lung carcinoma cells [27]. Curcumin increased the levels of E-cadherin and decreased the levels of Vimentin in HGF-stimulated cells. We investigated the levels of mesenchymal and epithelial proteins in HGF-stimulated cells. We observed that procaine suppressed the expression of HGF-induced mesenchymal markers while increasing epithelial markers.

The change in the cell morphology, alteration in the ability to migrate, and invasion are the hallmark features of EMT [38,39,40]. We demonstrated that procaine imparted HGF-induced invasion migration as well as metastasis. HGF treatment promoted cell invasion and migration, however, procaine significantly attenuated HGF-stimulated invasion and migration. MMP-2 and MMP-9 proteins are associated with the invasion, proliferation, and metastasis of cancer cells [41,42,43]. Recent studies have reported that HGF promotes the invasion of tumor cells associated with the up-regulation of MMP-2 and MMP-9 [44,45,46]. We noted that procaine suppressed the level of MMP-2 and MMP-9 proteins in HGF-stimulated cells, suggesting that the downregulation of invasion and metastasis by procaine could be due to the abrogation of expression of these proteins.

Although diverse pathways might affect EMT, activation of the HGF/c-Met pathway has been found to play a prominent role in the regulation of the EMT program in diverse cancers [26,47]. The deregulated activation of c-Met has been positively associated with metastasis in HCC [26]. It has been shown that c-Met is present in endothelial cells and that HGF can stimulate their growth and motility [48]. Moreover, the downregulation of c-Met activation can reduce the invasion and metastasis in various tumor cells [22,27,49,50]. Multiple reports have demonstrated that HGF/c-Met signaling cascades modulate the EMT process in various types of cancers [27,47,49,51,52]. Hence, we observed whether procaine could suppress the activation of the c-MET pathway. The data indicated that procaine could inhibit HGF-induced c-Met and its downstream pathways such as PI3K/Akt/mTOR, MEK/ERK, and JNK/p38.

Next, we observed that the treatment of procaine significantly reduced tumorigenesis and metastasis without notable toxicity in tested mice. Since procaine treatment up to 50 mg/kg did not display any major toxicity, we next demonstrated anti-tumor actions in the HCC model. Procaine had significant antitumor potential in orthotopic models at a dose of 50 mg/kg and significantly decreased lung metastasis. Additionally, we noted that procaine inhibited c-Met and its downstream signaling events such as PI3K/Akt/mTOR and MEK/ERK activation in tumor tissues. The modulation of expression of these proteins was consistent with the findings of our in vitro experiments.

## 4. Materials and Methods

### 4.1. Reagents

Procaine, 3-(4,5-dimethylthiazol-2-yl)-2,5-diphenyltetrazolium bromide (MTT) and bovine serum albumin (BSA) were purchased from Sigma-Aldrich (St. Louis, MO, USA). Alexa Fluor^®^ 488 donkey anti-mouse IgG (H+L) antibody and Fluor^®^ 594 donkey anti-rabbit IgG (H+L) antibody were obtained from Life Technologies (Grand Island, NY, USA). iN-fect™ in vitro Transfection Reagent was obtained from iNtRON Biotechnology (Seongnam, Korea). Antibodies against MnSOD, Fibronectin, Vimentin, E-cadherin, N-cadherin, Occludin, Twist, MMP-2, MMP-9, Akt, and β-actin were purchased from Santa Cruz Biotechnology (Santa Cruz, CA, USA). Antibodies against Snail, p-c-Met(Tyr1234/1236), c-Met, p-PI3K(Tyr458), PI3K, p-Akt(Ser473), p-mTOR(Ser2448), mTOR, p-MEK(Ser217/221), MEK, p-ERK(Thr202/Tyr204), and ERK were procured from Cell Signaling Technology (Beverly, MA, USA).

### 4.2. Cell Lines

Human HCC (HepG2 and HCCLM3) cells were purchased from the American Type Culture Collection (Manassas, VA, USA) and propagated in the cell culture medium supplemented with 10% FBS and 1% penicillin-streptomycin and were maintained at 37 °C under a 5% CO_2_ atmosphere.

### 4.3. MTT Assay

Cell viability was measured using the MTT assay as described previously [53,54]. HepG2 and HCCLM3 cells were pre-exposed to various doses of procaine (0, 30, 50, 100, 200 μM) and treated with procaine + HGF (50 ng/mL) for a total 24 h. Subsequently, 30 μL of MTT solution (2 mg/mL) was added and incubated for 2 h followed by the addition of MTT lysis buffer (100 μL) to lysis MTT formazans. The absorbance of the colored solution was quantified at 570 nm by VARIOSKAN LUX (Thermo Fisher Scientific Inc., Waltham, MA, USA).

### 4.4. Western Blotting Analysis

A western blot was carried out as reported earlier [55,56]. For detection of expression of protein of interest, whole cell extracts were prepared using cell lysis reagent upon exposure to procaine and HGF. Total protein concentration was estimated by the Bradford assay from Bio-Rad (Hercules, CA, USA). The western blot analysis was then executed as elaborated earlier [57]. The membranes were then detected through a Davinch-Chemi Fluoro Imager using a chemiluminescence (ECL) (EZ-Western Lumi Femto, DOGEN, Seoul, Korea).

### 4.5. Immunocytochemistry

HepG2 and HCCLM3 cells were pre-treated with procaine (200 μM) for 2 h followed by treatment with procaine and HGF (50 ng/mL) for a total 24 h. Thereafter, cells were fixed with 4% paraformaldehyde (PFA) and immunocytochemistry was carried out as indicated in our earlier studies [58]. The fluorescence signals were detected by an Olympus FluoView FV1000 confocal microscope (Tokyo, Japan).

### 4.6. Boyden Chamber Assay

The invasive potential was determined using a 48-well micro chemotaxis Boyden chamber (Neuro Probe, Cabin John, MD, USA). HepG2 and HCCLM3 cells were propagated in a Boyden chamber containing polycarbonate membrane (8-mm pore size) which has been coated with matrigel. Thereafter, the Boyden chamber assay was carried out as reported in our prior reports [59,60]. The invading cells on the membranes were observed by a Nikon ECLIPSE Ts2 (Nikon corporation, Tokyo, Japan).

### 4.7. Wound Healing Assay

HepG2 and HCCLM3 cells were propagated in a 6-well plate in a medium devoid of serum. When the cell confluence reached approximately 80%, a wound healing assay was carried out as elaborated earlier to quantify the percentage of migrated cells [61,62]. The width of wound was measured at four different sites by a Nikon ECLIPSE Ts2 (Nikon corporation, Japan), and non-treated samples were used as controls.

### 4.8. Gelatin Zymography

The proteolytic activity of MMP-2 and MMP-9 were analyzed using gelatin zymography as described earlier [58]. HepG2 and HCCLM3 cells were treated with procaine and HGF, and the supernatant medium was collected and concentrated. The samples were separated on 10% SDS-PAGE gel containing 0.1% gelatin. Gels were washed with 2.5% Triton X-100 and incubated in a zymo-reaction buffer at 37 °C under 5% CO_2_ overnight. The gels were stained with Coomassie brilliant blue (7% glacial acetic acid, 40% methanol, and 0.25% Coomassie brilliant blue R250) and then destained until the band was observed.

### 4.9. Acute Toxicity Studies

The protocol of all the animal studies was approved by the SingHealth Institutional Animal Use and Care Committee (protocol number: 2013/SHS/870). To study any possible adverse effects of procaine on the non-diseased animals, NCr nude female mice (eight-week-old) were intraperitoneally administered with two doses of procaine (either 25 mg/kg or 50 mg/kg) or solvent (0.1% DMSO). The animals were continuously kept under observation for the appearance of any adverse symptoms including phenotypic and behavioral changes. Any alteration in the body weight, food consumption, and water intake were monitored routinely until day 8. On day 8, the whole blood was collected from the animals of each group by cardiac puncture and the functions of vital organs (such as liver and kidney functions) were examined by measuring the activity of marker enzymes in the serum.

### 4.10. Orthotopic Implantation of HCC in Nude Mice

The protocol of all the animal studies was approved by the SingHealth Institutional Animal Use and Care Committee (protocol number: 2013/SHS/870). To study the efficacy of procaine as an antitumor agent in a mice model, NCr nude female mice (eight-week-old) bearing an orthotopic tumor (HCCLM3-Luc-cells derived) were intraperitoneally administered with two doses of procaine (either 25 mg/kg or 50 mg/kg) or solvent (0.1% DMSO) twice a week for four weeks. The progression of the tumor was examined two times a week by measuring the bioluminescence signals. All of the animals were sacrificed at the end of the study by carbo-di-oxide inhalation. Tumor-bearing primary tissues (liver) and secondary tissues (lungs) were harvested, snap-frozen, and stored in a deep freezer (−80 °C) for further study.

### 4.11. Immunohistochemical Study

The tumors obtained from experimental animals were fixed with 10% neutral buffered formalin (BBC Biochemical, Mount Vernon, WA, USA), processed, and made into paraffin blocks. Subsequently, IHC analysis was carried out as reported in our earlier studies [63]. The images were captured by a Nikon ECLIPSE Ts2 (Nikon corporation, Japan).

### 4.12. Statistical Analysis

An unpaired t-test was performed for statistical comparisons among the groups, and a one-way ANOVA was carried out to find the overall difference among the groups. *p* < 0.05 was considered statistically significant (GraphPad Prism 5.0; GraphPad Software, CA).

## 5. Conclusions

Overall, our findings indicate that procaine exhibited substantial inhibitory action on the HGF-stimulated EMT process. We noted that procaine significantly attenuated invasion and migration through abrogating the EMT-related proteins, and these changes were linked with negative regulation of c-Met and c-Met downstream signals. In vivo findings have also established the inhibitory effect on the EMT process of procaine, which can be a potential candidate as an anti-metastatic drug.

## Figures and Tables

**Figure 1 cancers-14-04978-f001:**
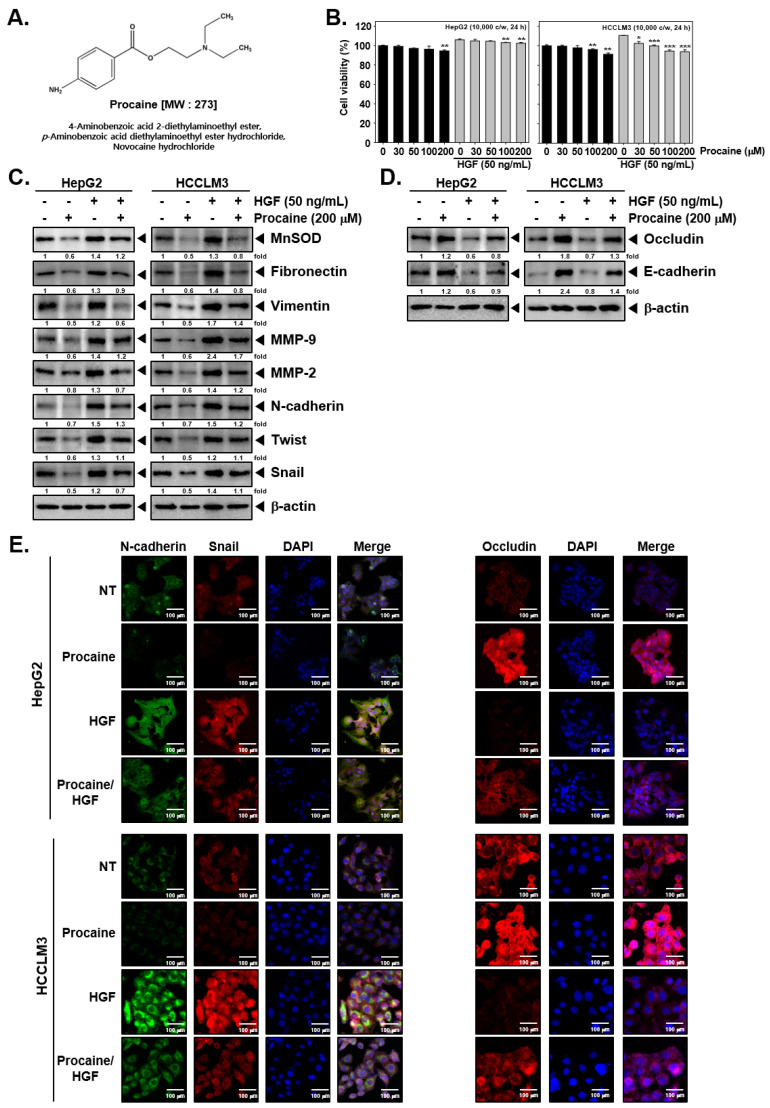
Impact of procaine on EMT markers. (**A**) The structure of procaine. (**B**) HepG2 and HCCLM3 cells were pre-treated with procaine (0, 30, 50, 100, 200 μM) for 2 h, followed by treated with procaine and HGF (50 ng/mL) for a total 24 h and cell viability was measured. Data represent means ± SD. * *p* < 0.05 vs. non-treated (NT) cells, ** *p* < 0.01 vs. non-treated (NT) cells, *** *p* < 0.001 vs. non-treated (NT) cells. (**C**,**D**) HepG2 and HCCLM3 cells were pre-treated with procaine (200 μM) for 2 h and treated with procaine and HGF (50 ng/mL) for a total 24 h, and western blotting was performed. (**E**) HepG2 and HCCLM3 cells were exposed to HGF and/or procaine as indicated in panel **C**, and the expression of different EMT markers was determined by immunocytochemistry. The results shown are representative of the three independent experiments.

**Figure 2 cancers-14-04978-f002:**
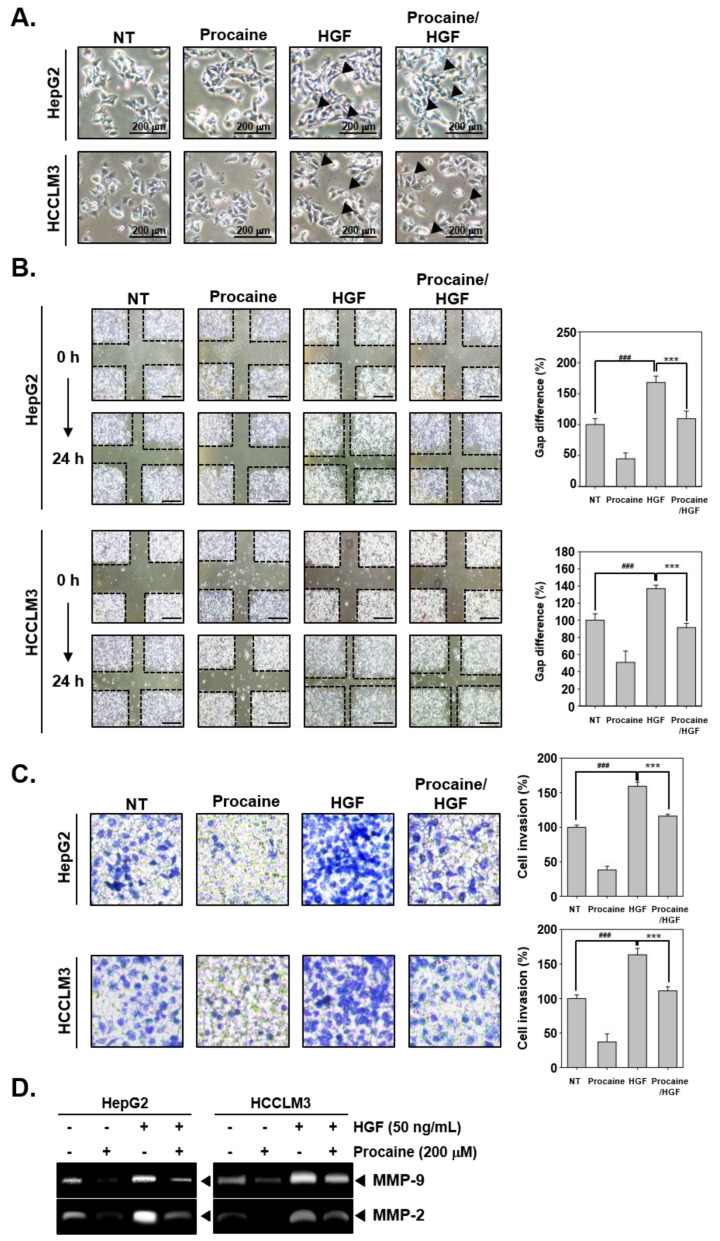
Influence of procaine on invasion and migration. (**A**) HepG2 and HCCLM3 cells were exposed to procaine (200 μM) or HGF (50 ng/mL) for 24 h. Morphological changes were then observed under the microscope. (**B**) HepG2 and HCCLM3 cells were exposed to procaine or HGF for 24 h, and then a wound healing assay was carried out (scale bar: 40X). The space between the lanes was measured at the beginning of the experiment and at 24 h of incubation. Data represents mean ± SD. ^###^ *p* < 0.001 vs. non-treated cells. (**C**) The invasiveness of HCC cells was measured using the Boyden chamber assay. Data represent mean ± SD. ^###^ *p* < 0.001 vs. non-treated cells and *** *p* <0.001 vs. HGF treated cells. The results shown are representative pictures of three independent experiments. (**D**) Gelatinolytic activity of MMP-9 and MMP-2 was determined by gelatin zymography.

**Figure 3 cancers-14-04978-f003:**
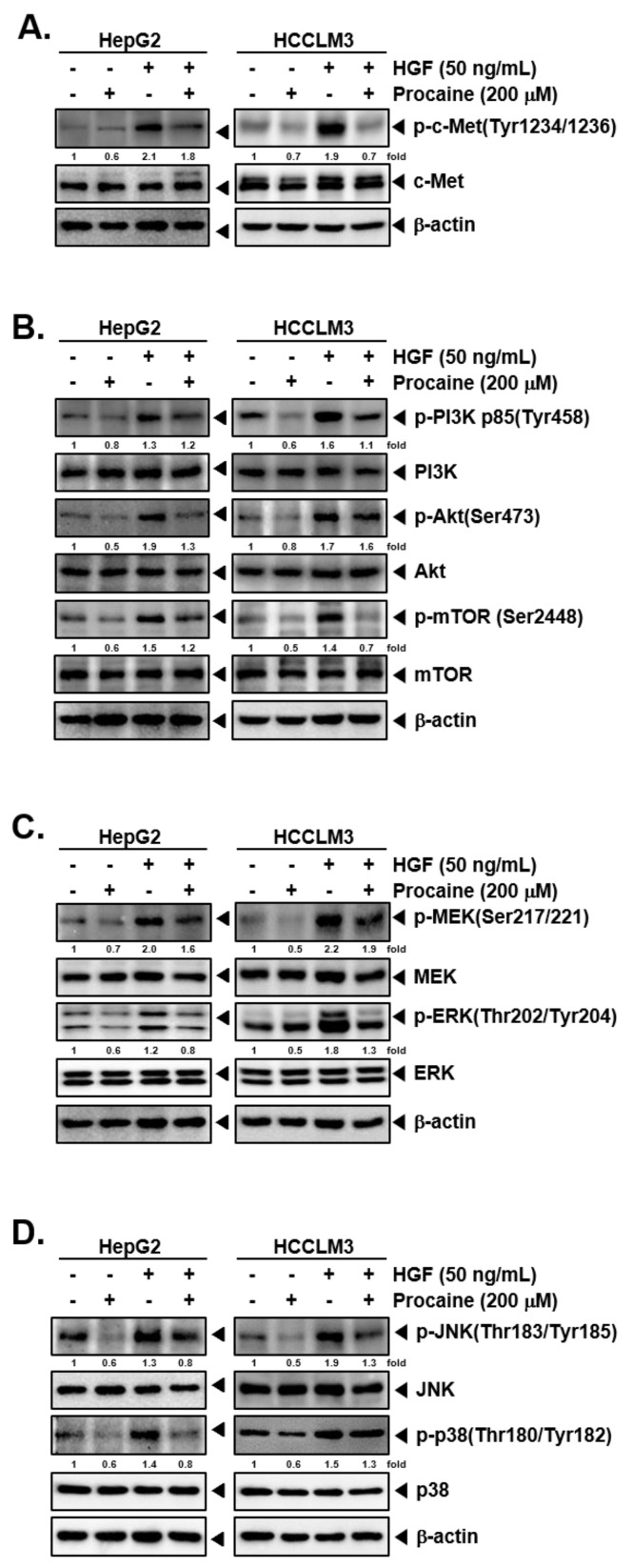
Influence of procaine on c-Met regulated signalling cascades. (**A**–**D**) HepG2 and HCCLM3 cells were pre-treated with procaine (200 μM) for 6 h, exposed to HGF (50 ng/mL) for 30 min, and subjected to western blotting.

**Figure 4 cancers-14-04978-f004:**
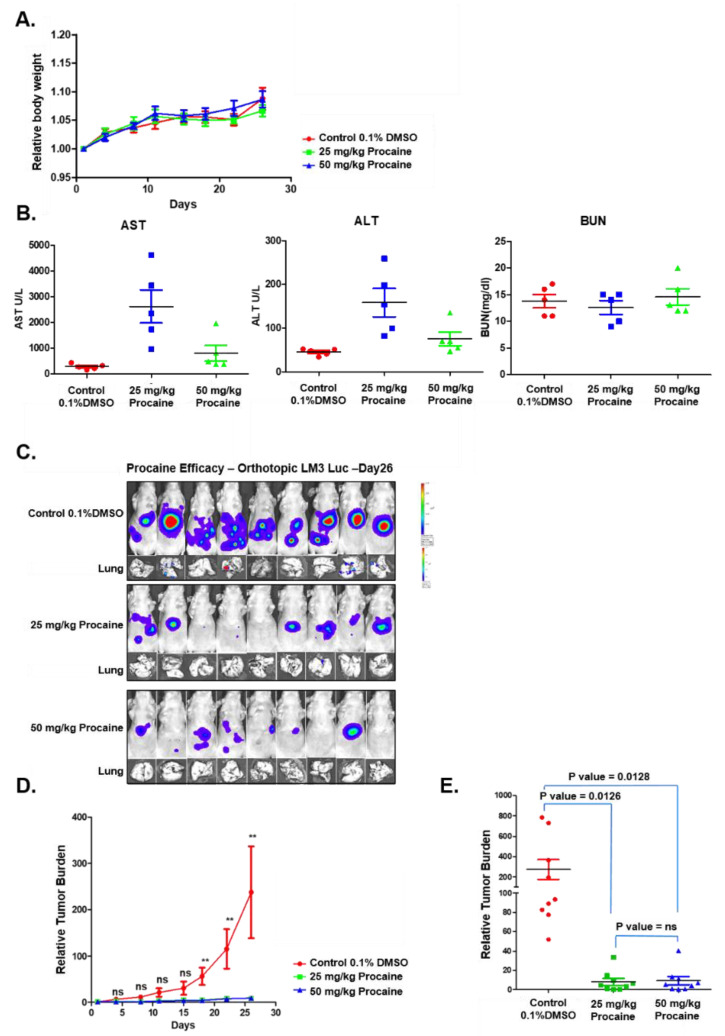
Effect of procaine on tumor growth. (**A**) The graph represents the body weight of mice throughout the acute toxicity study. (**B**) Levels of ALT and AST to analyze the liver function and BUN to determine the kidney function were measured in the serum of experimental animals. (**C**) HCCLM3-Luc cells-derived tumor tissues are orthotopically placed into the liver with a subsequent intraperitoneal administration of 0.1% DMSO or procaine (25 or 50 mg/kg) twice a week for four weeks. Lungs were examined for metastasis. Tumor growth and metastasis were monitored using bioluminescence imaging. (**D**) Tumor burden was measured in vehicle-treated or procaine-treated tumor-bearing mice. (** *p* < 0.01) (**E**) The scattered plot indicates tumor burden, which was measured in the different animals in control and at the different doses of procaine.

**Figure 5 cancers-14-04978-f005:**
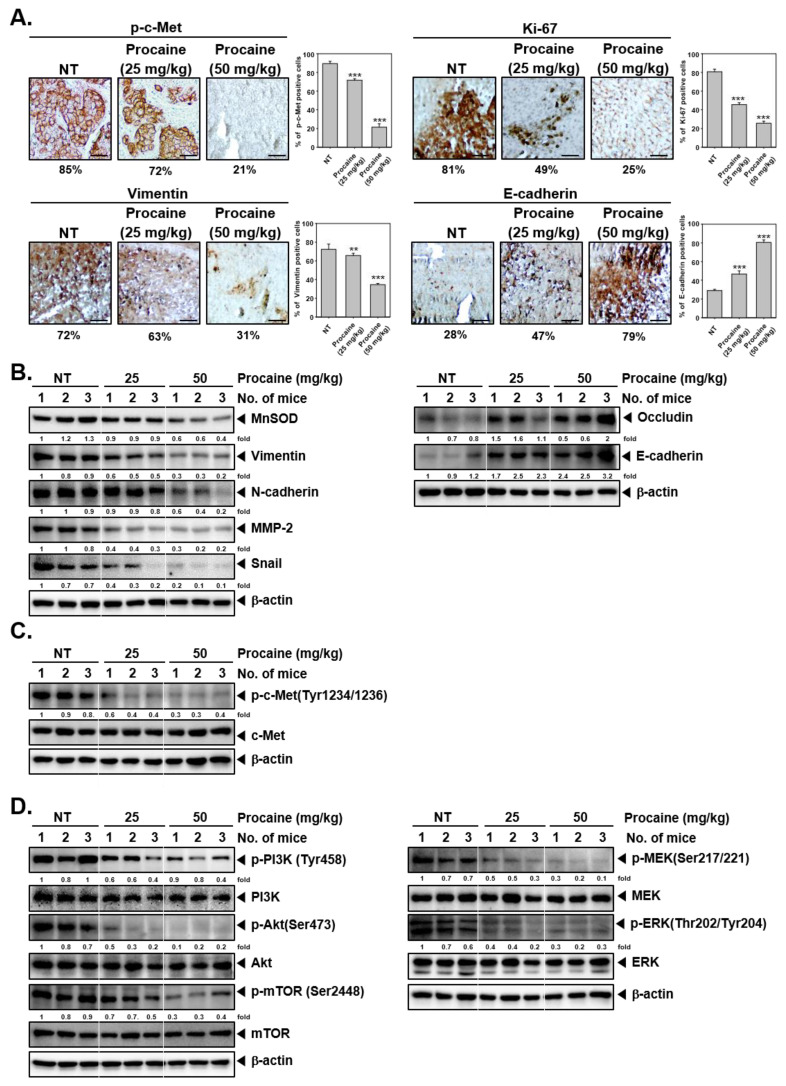
Impact of procaine on EMT and c-Met dependent signaling cascades in tumor tissues. (**A**) Examination of EMT markers and phospho-c-Met by IHC. The expression of phospho-c-Met, Ki-67, Vimentin, and E-cadherin was quantified and represented as mean ± SD on the right panel (scale bar: 200X). ** *p* < 0.01 vs. non-treated (NT) cells and *** *p* < 0.001 vs. non-treated (NT) cells. (**B**) The expression of EMT markers is analyzed using western blot analysis. (**C**,**D**) The expression of p-c-Met (Tyr1234/1236), c-Met, p-PI3K (Tyr458), PI3K, p-Akt (Ser473), Akt, p-mTOR (Ser2448), mTOR, p-MEK (Ser217/221), MEK, p-ERK (Thr202/Tyr204), and ERK was checked in tumor tissues.

## Data Availability

The data presented in this study are available on request from the corresponding author.

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
