# Peer review of "Procaine Abrogates the Epithelial-Mesenchymal Transition Process through Modulating c-Met Phosphorylation in Hepatocellular Carcinoma"

_cancers, 2022, doi:10.3390/cancers14204978_

Round 1

Reviewer 1 Report

The proposed manuscript provides complex, interesting and new data regarding the suppressive effect of procaine on tumor metastasis molecular mechanisms in hepatocellular carcinoma (HCC). This is an area of great interest since procaine displayed in recent studies different anti-cancer activities. The author used as cellular models for in vitro hepatic cancer the HepG2 cells (as cell line exhibiting epithelial-like morphology), and HCCLM3 cells (as cells with high metastatic potential) to evaluate specific biomarkers for the c-Met signaling pathways driven epithelial- mesenchymal transition (EMT)– a crucial step in cancer progression.

1.      In the Introduction section, instead of the concluding remarks: “We noted that procaine abrogated the EMT process by modulating the c-Met signaling pathway and suppressed cancer cell mobility. Procaine also repressed the growth of HCC tumors in the mice model and hampered lung metastasis.” the authors should mention, justify and explain the selection and the use of their experimental and preclinical models – the cultured cells (HepG2 and HCCLM3) and the tumor orthotopic cancer model.

2.      Like many pharmacologic active molecules, procaine exhibits a multimodal dose response. Although were initially tested a range of procaine concentrations between 30 – 200 μM, in all the experiments on cultured cells was used the same concentration of procaine: 200 μM. Could the authors also show/explain why this only higher dose was selected, as only one dose of procaine seems to displays effectiveness? As shown in Reference No 28, in most cancer experimental studies the procaine concentrations used were generally lower than 100 μM, and even lower than 30 μM.

3.      The figure 1.B. showing the effect of procaine versus HGF/procaine on HepCLM3 cells on cell viability deserves more explanations and analysis of the results and of statistical significance between procaine and HGF/procaine, between Control (0) and different concentrations.

4.      In the section 4.5. Immunocytochemistry, “HepG2 and HCCLM3 cells were incubated with procaine (200 mM) for 2 h followed by treatment with HGF (50 ng/ml) for 24 h”. The question is if HGF was added in the cell culture milieu or after procaine removal. This procedure could justify the use of the highest concentrations of procaine.

5.      In the figure 2 A and B the quality of the microscopy images should be improved if possible, and the “spindle-like” shape of the cells could be indicated with arrows.

6.      6. In the section 2.4. (page 6) the acute toxicity studies should be mentioned before the chronic studies.

7.      Although in figure 1 is depicted the chemical structure of procaine hydrochloride, the authors should mention the form of procaine used in experiments. Procaine hydrochloride is water soluble, however in the figure 4, in “in vivo” animal treatments, the control group received DMSO.

Author Response

Reviewer #1

The proposed manuscript provides complex, interesting and new data regarding the suppressive effect of procaine on tumor metastasis molecular mechanisms in hepatocellular carcinoma (HCC). This is an area of great interest since procaine displayed in recent studies different anti-cancer activities. The author used as cellular models for in vitro hepatic cancer the HepG2 cells (as cell line exhibiting epithelial-like morphology), and HCCLM3 cells (as cells with high metastatic potential) to evaluate specific biomarkers for the c-Met signaling pathways driven epithelial- mesenchymal transition (EMT)– a crucial step in cancer progression.

Comment 1. In the Introduction section, instead of the concluding remarks: “We noted that procaine abrogated the EMT process by modulating the c-Met signaling pathway and suppressed cancer cell mobility. Procaine also repressed the growth of HCC tumors in the mice model and hampered lung metastasis.” the authors should mention, justify and explain the selection and the use of their experimental and preclinical models – the cultured cells (HepG2 and HCCLM3) and the tumor orthotopic cancer model.

 Response: We have indicated about the cell lines and animal model used in the introduction section as you recommended. Thank you for your advice.

Comment 2. Like many pharmacologic active molecules, procaine exhibits a multimodal dose response. Although were initially tested a range of procaine concentrations between 30 – 200 μM, in all the experiments on cultured cells was used the same concentration of procaine: 200 μM. Could the authors also show/explain why this only higher dose was selected, as only one dose of procaine seems to displays effectiveness? As shown in Reference No 28, in most cancer experimental studies the procaine concentrations used were generally lower than 100 μM, and even lower than 30 μM.

Response: We have performed Western blot analysis for EMT-related proteins and result was shown in supplementary figure 1A. MnSOD and other proteins were markedly suppressed at 200 mM of procaine treated cells. Therefore, we selected the concentration of procaine at 200 mM. Thank you for your advice.

Comment 3. The figure 1.B. showing the effect of procaine versus HGF/procaine on HepCLM3 cells on cell viability deserves more explanations and analysis of the results and of statistical significance between procaine and HGF/procaine, between Control (0) and different concentrations.

Response: We have performed MTT assay for 24 h and data was shown in Fig. 1B. We have also added more explanations in result section. Thanks for your advice.

Comment 4. In the section 4.5. Immunocytochemistry, “HepG2 and HCCLM3 cells were incubated with procaine (200 mM) for 2 h followed by treatment with HGF (50 ng/ml) for 24 h”. The question is if HGF was added in the cell culture milieu or after procaine removal. This procedure could justify the use of the highest concentrations of procaine.

Response: We have added the HGF in the cell culture milieu without procaine removal. We added the more explanation in section 4.5. As shown in Fig.1B, HepG2 and HCCLM3 cells did not display significant cytotoxicity up to 200 mM.

Comment 5. In the figure 2 A and B the quality of the microscopy images should be improved if possible, and the “spindle-like” shape of the cells could be indicated with arrows.

Response: We have improved the images and added the arrows as you recommended. Thank you.

Comment 6.  In the section 2.4. (page 6) the acute toxicity studies should be mentioned before the chronic studies.

Response: We have changed the figure and result as you have recommended. Thank you.

Comment 7. Although in figure 1 is depicted the chemical structure of procaine hydrochloride, the authors should mention the form of procaine used in experiments. Procaine hydrochloride is water soluble, however in the figure 4, in “in vivo” animal treatments, the control group received DMSO.

Response: We checked and modified the structure. Thank you.

Reviewer 2 Report

The manuscript entitled “Procaine abrogates the epithelial-mesenchymal transition process through modulating c-Met phosphorylation in hepatocellular carcinoma” by Yang et al. studies different markers involved in EMT process in HCC. The aim of the manuscript is to demonstrate the role of procaine as an inhibitor of the EMT process and of its potential as an antimetastatic drug. 

The manuscript cites part of the literature on procaine effects but mostly from revues. A full part of the literature on similar drugs (lidocaine) are missing, some concerning the manuscript results like effects on HCC and lung cancer. Concerning the procaine itself, the apoptotic effect and its role on proliferation and migration are not thorougly presented (see Ying et al, 2017 as an example). Moreover, the authors state as “recent” papers dated 2007 but miss other publications that are more recent.

The question raised here is very interesting as there has not been a previous study on the role of Procaine and EMT progression. Nonetheless, the work is too preliminary and further investigation is required for this work to be suitable for publication.

Major concerns:

1.     Results and control experiments.

a.     Given the effects of procaine on cell viability, a proper experiment on cytotoxicity is lacking. The Fig1B compares procaine versus procaine+HGF but the propoer control would be with non-treated cells. Moreover, the cells were treated for 2h only with procaine and 48h with HGF. This protocol mainly reveals the effect of HGF on the cells as there is no indication on the half-life of procaine.

b.     In Fig1 different protocols were used in the different experiments (viability, WB and IF). The rationale behind these should be presented specially with the concern on apoptosis cited in 1a.

c.     Fig2a. focuses on the cell morphology. The resolution of the image is too low to observe the differences cited by the authors. A quantification and magnification of the phenotype is required.

d.     Fig2b and c focus on migration and invasion phenotypes. The migration assay shows cells at very different densities at 0h. These density differences can impact on the migration of the cells and the wound created. The gap difference can be better assessed when the cells form a proper front layer which is not the case here (see NT and Procaine in HepG2 and all conditions in HCCLM3). This experiment should therefore be redone for better reliability of the results.

e.     Fig3 depicts the c-Met signaling. The data are promising and can be better commented concerning the differences between the 2 cell lines.

f.      Concerning the toxicity in mice, the authors present a study for acute toxicity (Fig. 4E). What would the toxicity be with the protocole used in the study presented for tumorigenesis with a treatment with two doses of procaine, twice a week, for four weeks? Are there concerns on neurotoxicity of procaine as mentioned elsewhere?

Minor

-       No indication of the protein size on WBs. This would be very helpful regarding all blots: the activated versus non activated forms and the processed forms of certain proteins (MMPs).

-       The data of procaine on tumor growth (or apoptosis?) presented Fig4 are convincing. A comment on the increase of the levels of AST and ALT with 25 mg/kg of Procaine is required. How can this be explained as there seems to be less affect with higher amounts of the drug (50 mg/kg of Procaine)?

Author Response

The manuscript entitled “Procaine abrogates the epithelial-mesenchymal transition process through modulating c-Met phosphorylation in hepatocellular carcinoma” by Yang et al. studies different markers involved in EMT process in HCC. The aim of the manuscript is to demonstrate the role of procaine as an inhibitor of the EMT process and of its potential as an antimetastatic drug.

The manuscript cites part of the literature on procaine effects but mostly from revues. A full part of the literature on similar drugs

(lidocaine) are missing, some concerning the manuscript results like effects on HCC and lung cancer. Concerning the procaine itself, the apoptotic effect and its role on proliferation and migration are not thorougly presented (see Ying et al, 2017 as an example). Moreover, the authors state as “recent” papers dated 2007 but miss other publications that are more recent.

Response: The introduction section has been revised to include recent articles related to anticancer effects of procaine as suggested. Thanks.

The question raised here is very interesting as there has not been a previous study on the role of Procaine and EMT progression. Nonetheless, the work is too preliminary and further investigation is required for this work to be suitable for publication.

Major concerns:

  1. Results and control experiments.

Comment 1. Given the effects of procaine on cell viability, a proper experiment on cytotoxicity is lacking. The Fig1B compares procaine versus procaine+HGF but the propoer control would be with non-treated cells. Moreover, the cells were treated for 2h only with procaine and 48h with HGF. This protocol mainly reveals the effect of HGF on the cells as there is no indication on the half-life of procaine.

Response: We have performed MTT assay for 24 h and data has been shown in Fig. 1B. We have also added more explanations in result section. Thank you for your advice.

Comment 2. In Fig1 different protocols were used in the different experiments (viability, WB and IF). The rationale behind these should be presented specially with the concern on apoptosis cited in 1a.

Response: MTT assay was performed again in the same time conditions of WB and result was changed in Fig.1B as per your recommendation. Thank you.

Comment 3. Fig2a. focuses on the cell morphology. The resolution of the image is too low to observe the differences cited by the authors. A quantification and magnification of the phenotype is required.

Response: We have improved the images and added the arrows. Thank you for your advice.

Comment 4. Fig2b and c focus on migration and invasion phenotypes. The migration assay shows cells at very different densities at 0h. These density differences can impact on the migration of the cells and the wound created. The gap difference can be better assessed when the cells form a proper front layer which is not the case here (see NT and Procaine in HepG2 and all conditions in HCCLM3). This experiment should therefore be redone for better reliability of the results.

Response: We have performed wound healing assay again and replaced the result in Fig.2B. Thank you for your advice.

Comment 5. Fig3 depicts the c-Met signaling. The data are promising and can be better commented concerning the differences between the 2 cell lines.

Response: We added the comparison about efficacy of procaine between HepG2 and HCCLM3 cells in result section as you recommended. Thank you.

Comment 5. Concerning the toxicity in mice, the authors present a study for acute toxicity (Fig. 4E). What would the toxicity be with the protocol used in the study presented for tumorigenesis with a treatment with two doses of procaine, twice a week, for four weeks? Are there concerns on neurotoxicity of procaine as mentioned elsewhere?

Response: We did not investigate it as suggested but acute toxicity also examined other symptoms like changes in physical appearance, hunched back, increased respiration, arching and rolling, muscle spasm, tremors, cyanosis, stimulation or depression and no changes were noted.

Minor

Comment 6. No indication of the protein size on WBs. This would be very helpful regarding all blots: the activated versus non activated forms and the processed forms of certain proteins (MMPs).

Response: We have added the protein size for WBs in supplementary figure. Thank you.

Comment 7. The data of procaine on tumor growth (or apoptosis?) presented Fig4 are convincing. A comment on the increase of the levels of AST and ALT with 25 mg/kg of Procaine is required. How can this be explained as there seems to be less affect with higher amounts of the drug (50 mg/kg of Procaine)?

Response: We agree but at present do not have valid explanation for this observation, as it might require long-term toxicity analysis upon procaine treatment. Thanks.

Reviewer 3 Report

This MS presents procaine attenuated invasion and migration through abrogating the EMT process and c-Met-dependent signaling. Additionally, procaine reduced the tumor burden in an orthotopic mouse model and abrogated lung metastasis. I believe it could be published after major revision:

1.       When studying the effect of procaine on the cytotoxic activity, the EMT process related proteins and the expression of related proteins downstream of c-Met, the drug action time is different, which are 48h, 24h and 6h, respectively. What is the basis for selecting different action time?

2.       Procaine was administered at a dose of 25 mg / kg or 50 mg / kg for 4 weeks, while acute toxicity was a single dose. Whether the acute toxicity effect of a larger dose was investigated?

3.       Procaine is one of the frequently administered local anesthetic drugs. The acute toxicity only examines its impression on body weight and some biochemical indicators, and whether it has an impact on the exercise ability of mouse?

4.       Please describe the instruments used in the experiment.

5.     Please describe the detection method or instrument of biochemical indicators.

Author Response

This MS presents procaine attenuated invasion and migration through abrogating the EMT process and c-Met-dependent signaling. Additionally, procaine reduced the tumor burden in an orthotopic mouse model and abrogated lung metastasis. I believe it could be published after major revision:

Comment 1. When studying the effect of procaine on the cytotoxic activity, the EMT process related proteins and the expression of related proteins downstream of c-Met, the drug action time is different, which are 48h, 24h and 6h, respectively. What is the basis for selecting different action time?

Response: Since EMT occurs after the c-Met signaling pathway is activated, the experiment related to the c-Met signaling pathway was performed at earlier time points than the experiment to check the EMT-related genes to show that suppression of c-met pathway precedes the effect of procaine on EMT. In addition, MTT assay was performed again in the same time conditions of the experiment regarding EMT-related factors and result was changed in Fig.1B. Thank you.

Comment 2. Procaine was administered at a dose of 25 mg / kg or 50 mg / kg for 4 weeks, while acute toxicity was a single dose. Whether the acute toxicity effect of a larger dose was investigated?

Response: Yes, it was investigated with larger dose of 200mg/kg, in which 60% of mice died within few minutes after administration of the drug.

Comment 3. Procaine is one of the frequently administered local anesthetic drugs. The acute toxicity only examines its impression on body weight and some biochemical indicators, and whether it has an impact on the exercise ability of mouse?

Response: No, it does not have any impact on the exercise ability of mouse. The acute toxicity also examines other symptoms like changes in physical appearance, hunched back, increased respiration, arching and rolling, muscle spasm, tremors, cyanosis, stimulation or depression.

Comment 4. Please describe the instruments used in the experiment.

Response: We have added the information of instruments in materials and methods section. Thank you for your advice.

Comment 5. Please describe the detection method or instrument of biochemical indicators.

Response: We have added the information in materials and methods section as you recommended. Thank you.

Round 2

Reviewer 3 Report

 The research has certain practical application value and can be accepted after modification.